# Towards Trustworthy Automatic Diagnosis Systems by Emulating Doctors' Reasoning with Deep Reinforcement Learning

**Arsène Fansi Tchango** [†]
arsene.fansi.tchango@mila.quebec

**Rishab Goel** [‡]
rgoel0112@gmail.com

**Julien Martel** [§]
julien@datadoc.ca

**Zhi Wen** [†]   **Gaétan Marceau Caron** [†]   **Joumana Ghosn** [†]
{zhi.wen,gaetan.marceau.caron,joumana.ghosn}@mila.quebec
[†] Mila-Quebec AI Institute
[‡] Work done while at Mila-Quebec AI Institute
[§] Work done while at Dialogue Health Technologies Inc.

## Abstract

The automation of the *medical evidence acquisition* and *diagnosis process* has recently attracted increasing attention in order to reduce the workload of doctors and democratize access to medical care. However, most works proposed in the machine learning literature focus solely on improving the prediction accuracy of a patient's pathology. We argue that this objective is insufficient to ensure doctors' acceptability of such systems. In their initial interaction with patients, doctors do not only focus on identifying the pathology a patient is suffering from; they instead generate a differential diagnosis (in the form of a short list of plausible diseases) because the medical evidence collected from patients is often insufficient to establish a final diagnosis. Moreover, doctors explicitly explore severe pathologies before potentially ruling them out from the differential, especially in acute care settings. Finally, for doctors to trust a system's recommendations, they need to understand how the gathered evidences led to the predicted diseases. In particular, interactions between a system and a patient need to emulate the reasoning of doctors. We therefore propose to model the evidence acquisition and automatic diagnosis tasks using a deep reinforcement learning framework that considers three essential aspects of a doctor's reasoning, namely generating a *differential diagnosis* using an *exploration-confirmation* approach while prioritizing *severe pathologies*. We propose metrics for evaluating interaction quality based on these three aspects. We show that our approach performs better than existing models while maintaining competitive pathology prediction accuracy.

## 1   Introduction

In recent years, the digital healthcare industry has grown rapidly, benefiting from advances in machine learning (Esteva et al., 2019; Xiao et al., 2018). In particular, telemedicine, i.e., healthcare services provided via digital means, has received much attention (Kichloo et al., 2020). Aiming to reduce the workload of human doctors and thereby broaden access to care by automating parts of the interaction with patients, telemedicine applications typically involve two significant components, among others: *evidence acquisition* and *automatic diagnosis*. In a typical interaction, a patient first

36th Conference on Neural Information Processing Systems (NeurIPS 2022).

presents his/her chief complaint[1] to the telemedicine system, then the system asks further questions to gather more evidences (i.e., information about additional symptoms the patient might be experiencing and antecedents / risk factors the patient might have), and finally makes a prediction regarding the underlying diseases based on all collected evidences. Importantly, a human doctor would typically review the entire interaction, including the collected evidences and the predicted diseases, before establishing a diagnosis and deciding on next steps (e.g., ordering additional tests, preparing a prescription, etc.).

There are many existing works in the machine learning literature that aim to improve the automated evidence acquisition and disease diagnosis steps, using Reinforcement Learning (RL) (Kao et al., 2018; Wei et al., 2018; Yuan and Yu, 2021), Bayesian networks (Guan and Baral, 2021; Liu et al., 2022), or supervised approaches (Luo et al., 2020; Chen et al., 2022). These methods are generally trained to collect relevant evidences and to predict the pathology the patient is suffering from, while minimizing the number of questions asked to the patient.

However, these works overlook several crucial aspects of medical history taking, which is the process used by a doctor to interact with a patient with the goal of building the patient's medical history. Medical history is a broad concept encompassing various kinds of information relevant to a patient's current concerns (Nichol et al., 2018). It includes past illnesses, current symptoms, demographics, etc. In taking medical history, a doctor organizes the questions asked to a patient under an *exploration-confirmation* framework. At any point during the interaction, the doctor has in their mind a *differential diagnosis*, corresponding to a list of diseases worth considering, which might be modified throughout the interaction, based on the answers provided by the patient. In forming the differential diagnosis, the doctor treats *severe pathologies* differently by prioritizing questions that could help ruling them out even if they are less likely. In a usual in-person clinical setting, a physical examination will then be conducted to seek specific physical signs and improve the sensitivity and specificity of the medical history. A visual examination and/or supervised self-examination can also be conducted in a virtual care setting. After the medical history and physical examination are completed, the physician will decide if additional tests are necessary to establish the final diagnosis. Finally, a care plan will be established with the patient and proper follow-up, if needed. We next provide more details about the main concepts.

**Differential diagnosis** During the interaction with a patient, a medical doctor considers a set of plausible diseases, known as the *differential diagnosis* or simply *differential*, which is refined throughout the interaction based on the information provided by the patient (Henderson et al., 2012; Guyatt et al., 2002; Rhoads et al., 2017). The final differential often contains several diseases because the patient's symptoms and antecedents are insufficient to pinpoint a single pathology. As a result, the doctor might order follow up exams such as imagery and blood works to collect additional information to refine the differential and identify the pathology the patient is suffering from. Prior work on automatic diagnosis primarily focused on predicting the ground truth pathology, i.e., the disease causing the patient's symptoms (Chen et al., 2022; Zhao et al., 2021; Yuan and Yu, 2021; Guan and Baral, 2021; Wei et al., 2018; Xu et al., 2019; Kao et al., 2018). Richens et al. (2020) considers the differential in their approach; however it is unclear whether all evidences are provided to the model at the same time or if the model is part of the acquisition process. Our approach considers the differential diagnosis as an essential part of the model. Considering differentials instead of a single pathology has the added benefit of accounting for uncertainty and errors inherent in the diagnosis, especially when decisions are made solely based on interactions with patients and without medical exams or tests. For this reason, being able to predict the differential rather than the ground truth pathology is an important part of gaining the trust of doctors in a model.

**Exploration-confirmation** In addition to considering differentials, inquiring about evidences in a manner similar to the way a doctor would engage with patients is another important factor in gaining the trust of doctors in automated systems, and this is largely overlooked in prior works as well. Doctors generally proceed according to a diagnostic framework that provides an organized and logical approach for building a differential. More specifically, the interactions conducted by a doctor generally consist of two distinct phases: *exploration* followed by *confirmation*. During exploration, the doctor mainly asks questions that keep multiple pathologies under evaluation. In

---

[1]"A chief complaint is a concise statement in English or other natural language of the symptoms that caused a patient to seek medical care." https://www.ncbi.nlm.nih.gov/pmc/articles/PMC7161385/

that phase, pathologies can be removed from or added to the differential depending on the patient's answers. During the confirmation phase, the doctor inquires about evidences that strengthen the actual differential they are considering (Mansoor, 2018; Richardson and Wilson, 2015; Rhoads et al., 2017). This two-phase property of the interaction of doctors with patients is a separate dimension of the acquisition process that differs from the objective of predicting differentials, and should therefore be considered and measured separately from the common paradigm of optimizing prediction accuracy.

**Severe pathologies**    Among all possible diseases, severe pathologies receive more attention than others. Generally, a doctor does not want to miss out on them and would therefore explicitly gather evidences to rule them out from the differential as soon as they become plausible, even if they are less unlikely than other diseases (Rhoads et al., 2017; Ramanayake and Basnayake, 2018). Emulating this behavior within automated systems would contribute to increasing the trust of doctors in these systems.

In this work, we make a case for explicitly modelling the reasoning of doctors in designing evidence acquisition and automated diagnosis models using RL. More specifically, we focus on (1) using the **differential diagnosis**, rather than the ground truth pathology, as the training target of models, (2) modulating the evidence acquisition process to mimic a doctor' **exploration-confirmation** approach, and (3) prioritizing **severe pathologies**.

In the remaining sections, we review existing works and their shortcomings (Section 2); we describe our method that improves upon prior works especially in mimicking the reasoning of doctors (Section 3); we provide empirical results demonstrating the benefits of the proposed approach (Sections 4 and 5); finally, we discuss the limitations (Section 6) before concluding this work (Section 7). Our main contributions are: (1) we reformulate the task of evidence acquisition and automated diagnosis by introducing doctors' trust as a desideratum, and argue for explicitly designing models towards this goal; (2) we propose a RL agent, CASANDE, that promotes the desired behavior in addition to accurate predictions, by means of predicting differentials and reward shaping; (3) we empirically show that existing strong models on this task, even with modifications, are not sufficient for the proposed goal of mimicking the reasoning of human doctors, and that CASANDE improves over existing models while being competitive on conventional metrics.

## 2    Related work

There have been a variety of prior works on evidence acquisition and automatic diagnosis. For example, Wei et al. (2018) proposes to use DQN (Mnih et al., 2015) to tackle this problem. Kao et al. (2018) additionally proposes to create a hierarchy of symptoms and diseases based on anatomical parts and train an agent using Hierarchical Reinforcement Learning (Sutton et al., 1999). Xu et al. (2019), Zhao et al. (2021), and Liu et al. (2022) propose to encode relations among different evidences and evidence-disease pairs to enhance the efficiency of an agent trained using DQN. Peng et al. (2018) uses a policy gradient algorithm (Williams, 1992) and relies on reward shaping functions (Ng et al., 1999; Wiewiora et al., 2003; Devlin and Kudenko, 2012) defined in the state space to favour positive evidence acquisition. Our proposed approach also makes use of reward shaping functions, but unlike the work done in (Peng et al., 2018), those functions are defined in the action space and are used to induce a bias in our agent reflecting the reasoning of doctors.

In the aforementioned approaches, the space of the evidences to be collected and the one of the diseases to be predicted are merged together. Janisch et al. (2019) considers dealing with those spaces separately and proposes an agent which has two branches, one that decides which evidence to inquire about, trained using RL, and one that predicts the disease, trained using supervised learning. Other approaches enhance the training of the evidence inquiry branch by using information from the classifier branch. For example, Kachuee et al. (2019) relies on Monte-Carlo dropout (Gal and Ghahramani, 2016) to estimate the certainty improvement from the classifier branch output and uses that information as an evidence acquisition reward to train the acquisition branch. Yuan and Yu (2021) proposes an adaptive method to align the tasks performed by the two branches. More specifically, the acquisition branch receives an extra reward of the change in entropy of the disease prediction, and the acquisition process ends when the entropy of the prediction is below a dynamically learned, disease-specific threshold. Our approach also uses the two-branches setting and designs a mechanism to inform the acquisition process when to stop based on information from the classifier branch. Unlike

Yuan and Yu (2021), we explicitly introduce a *stop action* whose Q-value is trained to replicate a reward derived from the classification branch.

The evidence acquisition branch can be trained with learning paradigms different from RL. Chen et al. (2022) proposes to model the evidence acquisition process as a sequential generation task and trains the agent in a similar manner to BERT's Masked Language Modelling (MLM) objective (Devlin et al., 2019). Similarly, Luo et al. (2020) relies on randomly generated trajectories to train a system to collect evidences from patients in a supervised way. Finally, Guan and Baral (2021) makes use of the Quick Medical Reference belief network (Miller et al., 1986), and applies a Bayesian experimental design (Chaloner and Verdinelli, 1995) in the inquiry phase while relying on Bayes rule to infer the corresponding disease distribution.

However, most of these approaches are primarily focused on predicting the single ground truth pathology, as opposed to the differential diagnosis in our work. Also, they have rarely made specific efforts to shape or guide the interaction to resemble how doctors would interact with patients. As a result, it is debatable how much doctors could trust such systems, even though some may perform well on benchmark datasets, therefore casting doubt on their applicability in real-world applications.

## 3 Method

Let $E$ and $D$ be the number of all evidences (i.e., symptoms and antecedents) and diseases under consideration. Evidences can be *binary* (e.g., *are you coughing?*), *categorical* (e.g., *what is the pain intensity on a scale of 1 to 10?*), or *multi-choice* (e.g., *where is your pain located?*). The task of automated evidence collection and diagnosis can be viewed as a sequential decision process where each interaction with a patient is formalized using a finite-horizon Markov Decision Process $\mathcal{M} = (\mathcal{S}, \mathcal{A}, P, R, \gamma, T)$ with a state space $\mathcal{S} = \mathcal{S}' \cup \{\mathbf{s}_\perp\}$ (with $\mathbf{s}_\perp$ being the terminal state), an action space $\mathcal{A}$, a dynamics $P$, a reward function $R$, a discount factor $\gamma$, and a maximum episode length $T$. A state $\mathbf{s} \in \mathcal{S}'$ encodes socio-demographic data regarding the patient (e.g., age, sex) as well as the evidences provided by the patient so far. $\mathcal{A}$ is defined as $\{1, \cdots, E, E+1\}$ where the first $E$ elements, referred to as *acquisition actions*, are used to inquire about corresponding evidences, and the last element is the *exit action* that is used to explicitly terminate the interaction. At any point in time, only actions not yet selected in the episode are available for future selection. $P$ is deterministic and updates the current state based on the current action and the patient response if the current turn is less than $T$, otherwise, the next state is set to $\mathbf{s}_\perp$. Finally, regarding $R$ which is defined on the space $\mathcal{S} \times \mathcal{A} \times \mathcal{S}$ like $P$, each acquisition action is characterized by an inquiry cost $r_i$, and depending on whether the underlying patient is experiencing the corresponding evidence, it may additionally incur a retrieval reward $r_p$ or a missing penalty $r_n$. Upon the termination of the interaction, a prediction regarding the patient's differential is made based on the collected evidences.

An overview of the proposed approach is depicted in Figure 1. Our model is made of two branches: an evidence acquisition branch in charge of the policy for interacting with patients and a classifier branch in charge of predicting the differential at each turn. Both the classifier and the policy networks rely on a latent representation computed by an encoder from the evidences collected so far from the patient. Importantly, we introduce a *reward shaping module* which leverages the output of the classifier to define auxiliary rewards that help induce in the agent, during training, the desired properties of doctor reasoning identified in Section 1 . More precisely, the *exploration* and *confirmation* rewards are designed to encourage the two-phase based interactions, whereas the *severity* reward is used to explicitly handle severe pathologies. Finally, the *classification* reward is used to train the agent to predict the right differential at the end of the interaction process. The proposed agent predicts at each turn $t$ the next action $\mathbf{a}_t$ as well as the belief $bel_t$ regarding the patient's differential from the current state $\mathbf{s}_t$. We rely on a DQN variant algorithm, Rainbow (Hessel et al., 2018) (without noisy networks Fortunato et al. (2018)), to search for the optimal policy and the objective of the proposed auxiliary rewards is to encourage trajectories similar to those of experienced medical practitioners. In the next sections, we thoroughly describe each of these auxiliary rewards.

### 3.1 Exploration reward: encouraging the evaluation of multiple pathologies

We want our agent to undertake actions that favour considering several pathologies in the differential diagnosis during the first phase of the interaction. In practice, this means that the agent's belief is constantly changing after inquiring about a new evidence. As more evidences are collected, this

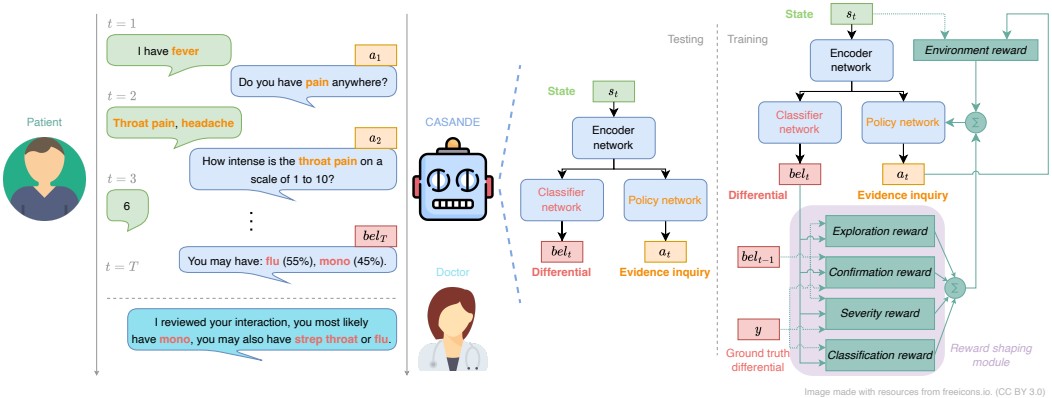

Figure 1: **Left**: A synthetic example of a patient's interaction with our proposed model. Importantly, the role of human doctors as the final decision maker is largely overlooked in prior works. The example is for illustration purposes only and is not meant as medical advice. **Middle**: The agent consists of two parts: an evidence acquisition module and a classifier responsible for generating the differential. **Right**: The reward shaping module leverages the output of the classifier to define auxiliary rewards that help induce desired features of the reasoning of medical doctors in the agent.

fluctuation should gradually reduce until it becomes insignificant. Based on this intuition, we derive the exploration reward $R_{\text{Ex}}$ as

$$R_{\text{Ex}}(\mathbf{s}_t, \mathbf{a}_t, \mathbf{s}_{t+1}) = \mathbb{1}_{\mathbf{s}_{t+1} \neq \mathbf{s}_\perp} w_{\text{Ex}}(t) \times JSD(bel_t, bel_{t+1}), \tag{1}$$

where $JSD$ is the Jensen-Shannon divergence that computes the dissimilarity between consecutive beliefs $bel_t$ and $bel_{t+1}$, and $w_{\text{Ex}}(t) \in [0, 1]$ is a dynamic weight which aims at controlling the importance of this shaping component as a function of time. Inspired from the shape of the *sigmoid* function, we design $w_{\text{Ex}}(t)$ as a translated version in $[0, T]$ of the "flipped" sigmoid function within the interval $[\bar{x}_{min}, \bar{x}_{max}]$. In other words, using $trans(t) = \frac{t*(\bar{x}_{max}-\bar{x}_{min})}{T} + \bar{x}_{min}$, we have

$$w_{\text{Ex}}(t) = sigmoid(-(trans(t) + \Delta_{\text{Ex}})), \tag{2}$$

where $\Delta_{\text{Ex}}$ is a hyper-parameter that controls how fast the function saturates at 0 (cf. Appendix A.1).

The beliefs $bel_t$ and $bel_{t+1}$ are generated by the classifier branch based respectively on states $s_t$ and $s_{t+1}$, with the latter state resulting from the execution of action $a_t$ when in state $s_t$.

### 3.2 Confirmation reward: Strengthening the agent's belief in the pathology

In the second phase of the interaction with the patient, the agent should inquire about evidences that help strengthen its belief regarding the differential diagnosis. This means that the more the agent collects evidences, the closer its belief should be to the ground truth distribution $y$. The importance of this shaping component should gradually increase as the agent moves towards the end of the interaction. From this intuition and inspired from potential-based reward shaping Ng et al. (1999), we derive the confirmation reward $R_{\text{Co}}$ as

$$R_{\text{Co}}(\mathbf{s}_t, \mathbf{a}_t, \mathbf{s}_{t+1}) = -\mathbb{1}_{\mathbf{s}_{t+1} \neq \mathbf{s}_\perp} w_{\text{Co}}(t) \times (\gamma\, CE(bel_{t+1}, y) - CE(bel_t, y)), \tag{3}$$

where $CE$ is the cross-entropy function, and $w_{\text{Co}}(t) \in [0, 1]$ controls the importance of this shaping component as a function of time. Like $w_{\text{Ex}}(t)$, $w_{\text{Co}}(t)$ is a translated version of the sigmoid function with the difference that it increases over time. Thus, we have:

$$w_{\text{Co}}(t) = sigmoid((trans(t) + \Delta_{\text{Co}})), \tag{4}$$

where $\Delta_{\text{Co}}$ is a hyper-parameter controlling how fast the function saturates at 1 (cf. Appendix A.1).

### 3.3 Severity reward: evidence gathering for ruling out severe pathologies

We want our agent to collect evidences that help rule out severe pathologies that are not part of the ground truth differential as soon as they become plausible. One proxy to achieve this for the agent is

to behave in such a way to monotonically increase the number $SevOut$ of severe pathologies that are not in both the ground truth differential and its predictions through time. As such, we define the severity reward $R_{\text{Sev}}$ as

$$R_{\text{Sev}}(\mathbf{s}_t, \mathbf{a}_t, \mathbf{s}_{t+1}) = \mathbb{1}_{\mathbf{s}_{t+1} \neq \mathbf{s}_\perp \& SevOut_{t+1} \neq SevOut_t}(\gamma SevOut_{t+1} - SevOut_t). \quad (5)$$

$SevOut_t$ and $SevOut_{t+1}$ correspond to the number of severe pathologies which are not in both the ground truth differential and the differentials respectively predicted at time $t$ and $t+1$, i.e., the beliefs $bel_t$ and $bel_{t+1}$. Here, a pathology is not part of a predicted distribution if its probability is below a given threshold.

### 3.4 Classification reward

This reward is designed to provide feedback to the agent describing how good its final predicted differential is with respect to the ground truth differential $y$ when the interaction process is over. Let $SevIn_t$ be the number of severe pathologies that are part of both the ground truth differential $y$ and the predicted belief $bel_t$, and let $Sev_y$ be the number of severe pathologies in $y$. As noted previously, a pathology is not part of a predicted distribution if its probability is below a given threshold. We define $\mathcal{V}(\mathbf{s}_t, y) = -CE(bel_t, y) + w_{\text{si}} \frac{SevIn_t}{Sev_y}$ as a measure of the quality of the belief prediction from state $\mathbf{s}_t$ where $CE$ is the cross-entropy function, and the second term, controlled by the hyper-parameter $w_{\text{si}}$, measures the inclusion rate of the relevant severe pathologies in $bel_t$. We thus define the classification reward $R_{\text{Cl}}$ as

$$R_{\text{Cl}}(\mathbf{s}_t, \mathbf{a}_t, \mathbf{s}_{t+1}) = \mathbb{1}_{\mathbf{s}_{t+1} = \mathbf{s}_\perp} \mathcal{V}(\mathbf{s}_t, y). \quad (6)$$

### 3.5 Training Method

The above mentioned auxiliary rewards are combined together as

$$F = \alpha_{\text{Ex}} R_{\text{Ex}} + \alpha_{\text{Co}} R_{\text{Co}} + \alpha_{\text{Sev}} R_{\text{Sev}} + \alpha_{\text{Cl}} R_{\text{Cl}}, \quad (7)$$

where the $\alpha$ weights are hyper-parameters, and the $(\mathbf{s}_t, \mathbf{a}_t, \mathbf{s}_{t+1})$ arguments for $F$ and all reward functions were dropped for clarity. The resulting function $F$ is then combined with the environment reward $R$, leading to a new reward function $R' = R + F$ that is used to train the policy network.

For simplicity, this section relies on DQN to present the loss for training each part of our agent, with the extension to Rainbow easy to do. Let $\theta$ and $\phi$ be the agent and target network parameters.

**Policy network loss function:**    The Q-value of the *exit* action does not depend on any following state. Also, the reward received when explicitly exiting is the classification reward. Therefore, to synchronize the agent branches, the *exit* action Q-value needs to be reconciled with the classification reward. This should be done at each interaction time step during the training, given that the agent needs to have a good estimate of the expected Q-value of the *exit* action to select this action. Thus,

$$Loss_Q(B) = \frac{1}{2} \underset{(\mathbf{s}_t, \mathbf{a}_t, r_t, \mathbf{s}_{t+1}, y) \in B}{\mathbb{E}} \left[ [Q_t - Q_\theta(\mathbf{s}_t, \mathbf{a}_t)]^2 + \mathbb{1}_{\mathbf{s}_{t+1} \neq \mathbf{s}_\perp} [Q_\theta(\mathbf{s}_t, \mathbf{a}_{exit}) - \mathcal{V}(\mathbf{s}_t, y)]^2 \right] \quad (8)$$

where $B$ is a batch from the replay buffer, $r_t = R(\mathbf{s}_t, \mathbf{a}_t, \mathbf{s}_{t+1})$ is the $R$ reward at time $t$, $Q_\theta(\mathbf{s}_t, \mathbf{a}_{exit})$ is the predicted Q-value for the pair $(\mathbf{s}_t, \mathbf{a}_{exit})$, $Q_t = r'_t + \mathbb{1}_{\mathbf{s}_{t+1} \neq \mathbf{s}_\perp} \gamma \max_{\mathbf{a} \in \mathcal{A}} Q_\phi(\mathbf{s}_{t+1}, \mathbf{a})$ is the target Q-value for the pair $(\mathbf{s}_t, \mathbf{a}_t)$ with $r'_t = R'(\mathbf{s}_t, \mathbf{a}_t, \mathbf{s}_{t+1}) = r_t + F(\mathbf{s}_t, \mathbf{a}_t, \mathbf{s}_{t+1})$.

**Classifier network loss function:**    The classifier is updated using the following loss at the end of an interaction:

$$Loss_C(B) = \frac{1}{2} \underset{(\mathbf{s}_t, \mathbf{a}_t, r_t, \mathbf{s}_{t+1}, y) \in B}{\mathbb{E}} \left[ \mathbb{1}_{\mathbf{s}_{t+1} = \mathbf{s}_\perp} CE(bel_t, y) \right]. \quad (9)$$

**Training process:**    Using the loss functions defined in Equations 8 and 9, the two branches are alternatively updated until the stopping criteria are satisfied. Both updates share the same batch of data $B$ which is sampled from the replay buffer. The training process is described in Appendix A.2.

# 4 Experiments

**Datasets** The datasets used in prior works such as DX (Wei et al., 2018), Muzhi (Xu et al., 2019), SymCAT (Peng et al., 2018), HPO (Guan and Baral, 2021), and MedlinePlus (Yuan and Yu, 2021) do not provide differential diagnosis information, and are therefore ineligible for validating the approach proposed in this work. Besides, as supported by Yuan and Yu (2021), the patients simulated using SymCAT are not sufficiently realistic for testing automatic diagnosis systems. We instead use the DDXPlus dataset (Fansi Tchango et al., 2022) for that purpose. In addition to providing differential diagnosis and pathology severity information, the DDXPlus dataset, unlike the above mentioned datasets, is not restricted to binary evidences, but also has categorical and multi-choice evidences, allowing for more efficient interactions with patients. Furthermore, the dataset contains 49 pathologies and 223 evidences (corresponding to 110 symptoms and 113 antecedents). Finally, the dataset is split into 3 subsets: a training subset with more than $10^6$ synthetic patients, and validation and test subsets containing roughly $1.4 \times 10^5$ synthetic patients each.

**Baselines** We consider 4 baselines, AARLC (short for Adaptive Alignment of Reinforcement Learning and Classification) (Yuan and Yu, 2021), Diaformer (Chen et al., 2022), BED (Guan and Baral, 2021), and BASD which is an extension on Luo et al. (2020). AARLC demonstrates SOTA results on the SymCAT dataset while Diaformer shows competitive results on the Muzhi and DX datasets. BED, an approach that does not require training, has impressive results on the HPO dataset. All these methods are designed to only handle binary evidences and had to be modified to deal with the different evidence types in the DDXPlus dataset. Moreover, those methods were designed to predict the ground truth pathology, and had to be modified to handle differentials. Additional details are provided in Appendix B.

**Experimental settings** Each patient in the DDXPlus dataset is characterized by a chief complaint, which is presented to the models at the beginning of the interaction, a set of additional evidences that the models need to discover through inquiry, a differential diagnosis, and a ground truth pathology. We allow interactions to have a maximum of $T = 30$ turns. We tune the hyper-parameters for each model, including the baselines, separately on the validation set and use the resulting optimal set of parameters to report the performances. For more details, see Appendix C.

**Evaluation metrics** We report on the interaction length (IL), the recall of the inquired evidences (PER), the ground truth pathology accuracy when only considering the top entry of the differential (GTPA@1) and when considering all entries (GTPA), the F1 score of the predicted differential diagnosis (DDF1), and the harmonic mean of the rule-in and rule-out rates of severe pathologies (DSHM). See Appendix D for the definition of these metrics. The GTPA@1 metric is only relevant for models trained to predict the patient's ground truth pathology. It is not relevant for models trained to predict the differential as the ground truth pathology is not always the top pathology in the differential (Fansi Tchango et al., 2022). We do not report the precision of the inquired evidences as it can be useful and sometimes necessary to ask questions that lead to negative answers from patients. For example, such questions need to be asked in order to safely rule out severe pathologies. Measuring the quality of the collected negative evidence is left as future work.

# 5 Results

**Disease prediction and evidence acquisition** Table 1 depicts the performance of each model at the end of the interaction process. We first focus on metrics measuring the quality of the predicted diseases. All models are on par with regard to the recovery of the ground truth pathology (GTPA), even when they are trained to predict the differential and are not given any indication about the pathology in the ground truth differential that corresponds to the disease the patient is suffering from. As for recovering the right differential (DDF1), models perform poorly when trained to predict the ground truth pathology even if some of those models generate a posterior pathology distribution. Performance greatly improves when models are trained to predict the differential, with CASANDE doing significantly better than all other models, outperforming the second-best model by more than 10% absolute on average. Likewise, CASANDE outperforms the baselines in handling the severe pathologies (DSHM) with an absolute margin of more than 4% when trained with the differential.

Table 1: Evaluation performance on the test set. Values indicate the average of 3 runs, and numbers in brackets indicate 95% confidence intervals. Values in **bold** indicate the best performance for a column, and values in *italic* indicate those that are not statistically significantly worse than the best ($p > 0.05$). BED is deterministic and related results are from 1 run. ↓ indicates lower is better and ↑ indicates higher is better.

| | Method | IL ↓ | Values expressed in percentage (%) ↑ | | | | |
| | | | GTPA@1 | DDF1 | DSHM | PER | GTPA |
|---|---|---|---|---|---|---|---|
| Trained With Differential | **CASANDE** | 19.71 (0.46) | 69.7 (3.54) | **94.24 (0.55)** | **73.88 (0.34)** | **98.39 (0.86)** | 99.77 (0.03) |
| | **Diaformer** | 18.41 (0.07) | 73.62 (0.60) | 83.3 (2.10) | 69.32 (0.73) | 92.92 (0.30) | 99.01 (0.33) |
| | **AARLC** | 25.75 (2.75) | 75.39 (5.53) | 78.24 (6.82) | 69.43 (2.01) | 54.55 (14.73) | 99.92 (0.03) |
| | **BASD** | 17.86 (0.88) | 67.71 (1.19) | 83.69 (1.57) | 65.06 (2.30) | 88.18 (1.12) | 99.30 (0.27) |
| Trained Without Differential | **CASANDE** | 19.53 (0.87) | 98.8 (0.54) | 30.84 (0.32) | 10.62 (0.22) | *98.07 (1.91)* | 99.46 (0.46) |
| | **Diaformer** | 18.45 (0.33) | 91.81 (1.95) | 30.38 (14.72) | 19.90 (31.07) | 92.61 (1.12) | 96.28 (6.21) |
| | **AARLC** | 6.73 (1.35) | **99.21 (0.78)** | 31.28 (0.38) | 10.96 (0.26) | 32.78 (13.92) | **99.97 (0.01)** |
| | **BASD** | 17.99 (3.57) | 97.15 (1.70) | 31.31 (0.29) | 10.81 (0.29) | 88.45 (5.78) | 98.82 (1.03) |
| | **BED** | **5.47** | 99.47 | 31.01 | 10.78 | 18.62 | 99.76 |

Those results indicate that training on the differential is essential and that CASANDE significantly outperforms the baselines.

CASANDE achieves the best performance on the PER metric demonstrating its ability to collect relevant evidences. We observe that the PER score also improves for AARLC when trained on the differential, suggesting that differentials are beneficial for collecting evidences. This improvement is not observed for BASD because its differential classifier branch only operates at the end of the interaction process, after the evidence collection. Similarly, Diaformer generates the differential at the end of the evidence collection and doesn't benefit from it to improve the collection process.

Finally, CASANDE has the longest interaction length. This is unsurprising, as CASANDE is trained to emulate the reasoning of doctors. In particular, the exploration and confirmation phases may lead to the model asking more questions. Additionally, CASANDE's interaction length is still considerably lower than the maximal value (30), which indicates that CASANDE is capable of terminating an interaction rather than always continuing to the end. Together, this shows that the slight increase in CASANDE's interaction length (less than 2 turns compared to the model with the shortest length) may be due to the different priorities of CASANDE and the baselines.

A qualitative evaluation of CASANDE was performed by a doctor who analyzed CASANDE's predictions for 20 patients randomly selected from the test set. The profiles of those patients, the predictions of CASANDE as well as the doctor's evaluation are presented in Appendix F. The doctor concluded that the evidences collected by CASANDE are indeed helpful for establishing a differential. Meanwhile, he cannot definitively assess CASANDE's disease predictions since, unlike CASANDE, the scope of diseases he considers is not limited to the 49 pathologies in DDXPlus.

**Trajectory quality evaluation**     We plot in Figure 2 the *confirmation score* versus the *exploration score* throughout the trajectories for the different models. The exploration score captures how distant two consecutive agent predictions are, while the confirmation score captures how close the agent prediction is with respect to the target distribution (see Appendix D.2 for the definitions). Ideally, an agent would start with a high exploration score and a low confirmation score (upper-left corner), gradually decreases the former and increases the latter, until the exploration score reaches a low value and the confirmation score reaches a high value at the end of the interaction (lower-right corner). As can be observed, this trend is highly followed by CASANDE which exhibits the highest exploration score at the beginning of the interaction and the highest confirmation score at the end of the interaction while, unlike other models, it is consistently moving towards the lower-right corner of the chart. Appendix G presents the differentials predicted by CASANDE at each interaction turn for 3 patients from the test set. We clearly observe CASANDE exploring various differentials at the beginning of the interaction and then focusing on a differential towards the end of the interaction.

**Handling of severe pathologies**     We are interested in analysing the pace at which severe pathologies are ruled out from or ruled in within the predicted differentials throughout the trajectories. We show in Figure 3 the curves corresponding to the average values of the harmonic mean scores of the rule-in and rule-out rates (i.e., DSHM), the *rule-out* rate, and the *rule-in* rate of severe pathologies throughout the trajectories (see Appendix D.1 for the definitions). As observed, CASANDE improves significantly on the harmonic mean throughout a trajectory, and eventually performs the best among

all models. This shows that the capability of CASANDE for handling severe pathologies is heavily grounded on the evidence it gradually collects. This evidence-based behavior is further confirmed by the rule-out and rule-in rates as CASANDE is the only method that significantly improves on both rates throughout the trajectories.

Interestingly, when focusing on the rule-out curve, it is noticeable that CASANDE performs the second-worst at the beginning, trailing the best performing model by more than $20\%$, but it improves to the second-best at the end, with the gap smaller than $3\%$. This suggests that at the beginning, CASANDE is more lenient on including severe diseases in its differentials, resulting in more false positives than others. However, this is a desired behavior of CASANDE, as during the exploration phase, it is expected to consider unlikely but severe diseases before gathering enough evidences to rule them out. In other words, at the beginning of a trajectory where few evidences are gathered, a model should err on the side of caution by keeping unlikely but severe diseases into consideration.

**Ablation studies**  We perform an ablation analysis to illustrate the contribution of each auxiliary reward component of

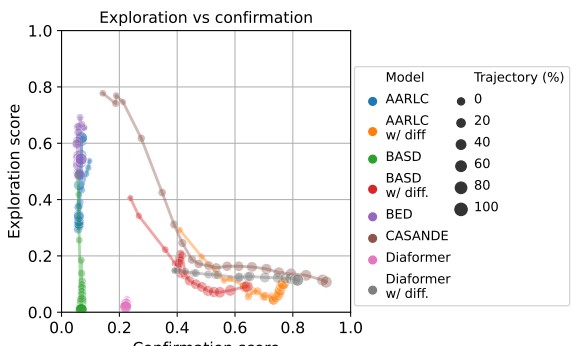

Figure 2: Confirmation score vs exploration score throughout trajectories on average for each model on the test set. For each trajectory, the scores are computed for each turn. We then select a fixed number of 21 equally spaced turns, which represent 0%, 5%, $\cdots$, 100% of the trajectory, and visualize the scores at these turns. Finally, we average these 21 turns across trajectories for each method.

CASANDE to the overall performance. Table 2 shows the results when some reward components are deactivated. We observe a drop in DDF1 and DSHM when most rewards are disabled. This indicates that each reward function helps the agent capture complementary information that is useful for predicting the differential. Also, as expected, the exploration and confirmation scores contribute in increasing the interaction length. Finally, we observe a decrease of the DSHM metric when the severity reward is not used. Additional studies are provided in Appendix E.

## 6  Limitations and potential negative social impact

**Limitations**  In this work, we set out to include doctors' trust as part of the desiderata in building evidence acquisition and automated diagnosis systems. In doing so, we focus our efforts on three essential aspects of the reasoning of doctors, namely the generation of a differential diagnosis, the exploration-confirmation approach, and the prioritized handling of severe pathologies. We chose to focus on them as advised by our collaborating physician. However, to most accurately evaluate doctors' trust, the proposed method has to be deployed so as to allow doctors different from the one

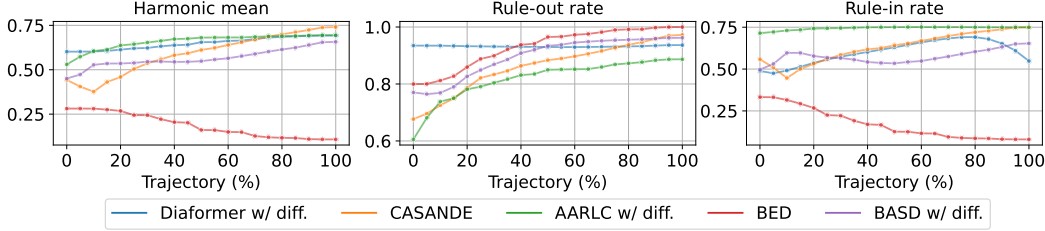

Figure 3: Average harmonic mean of the rule-out and rule-in rates, rule-out rate and rule-in rate on severe pathologies throughout trajectories. For each trajectory, their values are calculated at each turn. We then select a fixed number of 21 equally spaced turns, which represent 0%, 5%, $\cdots$, 100% of the trajectory, and visualize the scores at these turns. Finally, we average these 21 turns across trajectories for each method. The results are on the test set.

Table 2: Impact of the auxiliary reward components on the differential prediction and evidence collection, as measured on the validation set. All experiments are repeated 3 times with different random seeds. Values are in % except for IL. Values are expressed as an average $\pm$ a standard deviation. Values in brackets indicate 95% confidence intervals. Values in **bold** indicate the best performance for a column, and values in *italic* indicate those that are not statistically significantly worse than the best ($p > 0.05$).

| $R_{Ex}$ | $R_{Co}$ | $R_{Sev}$ | $R_{Cl}$ | IL | DDF1 | DSHM | PER |
|---|---|---|---|---|---|---|---|
| × | × | × | × | **17.56 (0.25) $\pm$ 0.06** | 93.11 (0.70) $\pm$ 0.16 | 73.55 (0.27) $\pm$ 0.06 | *98.24 (0.36) $\pm$ 0.08* |
| × | × | × | ✓ | 17.79 (0.13) $\pm$ 0.03 | 93.48 (1.31) $\pm$ 0.30 | 73.33 (0.28) $\pm$ 0.06 | *98.32 (0.88) $\pm$ 0.20* |
| ✓ | ✓ | ✓ | × | 19.09 (0.38) $\pm$ 0.09 | *94.26 (2.82) $\pm$ 0.65* | **74.19 (0.41) $\pm$ 0.10** | *98.34 (1.13) $\pm$ 0.26* |
| × | ✓ | ✓ | ✓ | 18.01 (0.36) $\pm$ 0.08 | **94.49 (0.27) $\pm$ 0.06** | *73.92 (0.38) $\pm$ 0.09* | *98.48 (1.10) $\pm$ 0.26* |
| ✓ | × | ✓ | ✓ | 18.91 (1.36) $\pm$ 0.32 | *94.16 (0.69) $\pm$ 0.16* | 73.86 (0.34) $\pm$ 0.08 | **98.54 (0.68) $\pm$ 0.16** |
| ✓ | ✓ | ✓ | ✓ | 19.71 (0.46) $\pm$ 0.11 | *94.24 (0.55) $\pm$ 0.13* | 73.88 (0.34) $\pm$ 0.08 | *98.39 (0.86) $\pm$ 0.20* |

we collaborated with to judge our approach. Additionally, it is reasonable to assume that doctors in other locations and/or specialties may have different ways of engaging with patients. Therefore, while we provide evidence in the medical literature (Section 1) to show the identified desiderata are widely applicable, we recognize that there may be cases where this work does not apply. In this work, we conduct experiments on synthetic patients, mainly due to the lack of real-patient datasets that contain differential diagnoses. We recognize that synthetic patients can be different from real patients in various and important ways, and that therefore results reported on synthetic patients may not extend to real patients.

**Potential negative social impact**   As we have made clear from the beginning, evidence acquisition and automated diagnosis systems are not substitutes for human doctors, but rather they are supportive tools for doctors, who should make the final decisions. However, the predictions of such systems potentially might be provided to patients as the final medical advice, without the intervention of human doctors. In such cases, the instructions that patients receive may be misleading or erroneous, and thus can do more harm than good to patients' health.

## 7   Conclusion

We reflected on the task formulation of evidence acquisition and automatic diagnosis that are essential for telemedicine services, and introduced doctors' trust as an additional desideratum. Concretely, we argued that emulating the reasoning of doctors is critical for gaining their trust, and we identified three essential doctor reasoning features that models can mimic. We proposed a novel RL agent with these features built in. We showed empirically existing models are insufficient for imitating the reasoning of doctors. We then demonstrated the importance of the explicit modelling of the differential diagnosis, and the efficacy of our model in emulating doctors, while being competitive on conventional metrics. This work is a first step towards reshaping the research in automatic diagnosis systems, and there is abundant potential for future work to explore. First, it is important to continue working with doctors to determine whether additional medical elements need to be considered when building machine learning approaches. Second, we need to build datasets that cover a wide spectrum of pathologies and train agents using this extensive diagnostic space along with all the corresponding evidences to uncover if the learned strategies are similar to expert doctors and how well machine learning approaches scale when the action space and the pathology space become much larger. Third, it is important to find a way of measuring the quality of collected negative evidences. Finally, it would be useful to consider online learning methods where doctors could identify missing evidences and give their overall feedback on the collected medical history.

## Acknowledgments and Disclosure of Funding

We would like to thank Dialogue Health Technologies Inc. for providing us access to the physician who supported us throughout this work and to some of the computational resources used to run the experiments. We would also like to thank Quebec's Ministry of Economy and Innovation and Invest AI for their financial support.

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
