# OpenReview forum: "Towards Trustworthy Automatic Diagnosis Systems by Emulating Doctors' Reasoning  with Deep Reinforcement Learning"
_NeurIPS.cc/2022/Conference — NeurIPS 2022 Accept_

### Official Review · Reviewer_CRMB · 2022-07-09

**Rating:** 6
**Confidence:** 3
**Soundness:** 3 good
**Presentation:** 4 excellent
**Contribution:** 4 excellent

**Summary:**

The paper proposes a novel RL architecture which is specially designed to help medical doctor’s diagnosis. The system utilizes differentiable diagnosis by incorporating a classification network, and considers three essential aspects of doctors’ reasoning, namely exploration, confirmation and severe pathologies. The system is shown to empirically work by experiments.



**Questions:**

1.DDXPlus dataset along is used. Although evaluated thoroughly, is there any special reason that you do not consider other dataset?
2.What’s the impact when one adjust the architecture of the classifier network?
3.The system depends essentially on the data quality. However, some patients may be tricking, so that the data quality of the patient is not good. Do you consider this case since I think this often happens? Like using an adversarial agent to deal with this?


**Limitations:**

Not obvious.

**Strengths And Weaknesses:**

Strength:
1.The paper considers a system which physically models the diagnosis process of the doctor.
2.Experimental results are promising
3.The presentation is clear
Weakness:
1.Theoretical contribution seems limited.

---

> ### Author Response · Authors · 2022-08-02
> **Feedback to reviewer CRMB**
>
> Weakness 1: Our paper is not a fundamental ML/RL research paper. It is an applied research paper describing an approach that is tailored to a medical problem, and that attempts to bridge the gap between the ML and medical communities as the problem definition that ML researchers are working on is incomplete / incorrect and doesn’t include several elements that are deemed important by the medical community. Please see the overall comment above.
>
> Question 1: We initially created a synthetic dataset based on the publicly available SymCat medical knowledge base, which is used in several deep learning / reinforcement learning papers, and we trained our first RL agent with this dataset. We then asked a doctor to verify the quality of this dataset. The doctor pointed out several problems: 1) lack of differential diagnosis, 2) no indication of the severity of pathologies, 3) some unrealistic patients which we traced back to incorrect stats / probabilities in SymCat, and 4) some problematic symptom definitions. We decided to drop this dataset because it couldn’t be used to handle several aspects deemed crucial by the doctor (such as the differential diagnosis and the pathology severity information).
> We also checked other publicly available datasets. Those datasets lack several of the important elements doctors care about. Moreover, the publicly available datasets that contain data about real patients are very small and incomplete: the DX dataset has 527 patients with only 5 pathologies; patients do not have a differential diagnosis and the diseases of each medical diagnosis conversation are labeled automatically by a medical dialogue system (instead of a human). The MuZhi dataset has 710 patients with only 4 pathologies and no differential diagnosis; this dataset also lacks conversations between the human doctors and patients. The DDXPlus dataset is the only dataset that we found which has the differential diagnosis and severity information, with a large number of patients, and a large set of pathologies and evidences (which is an important point if we want to verify how an RL agent behaves when the action space is large). Our collaborating doctor analyzed the DDXPlus dataset and was happier with the quality of this dataset than the SymCat-based dataset. But this doesn’t mean that the DDXPlus dataset is perfect; the doctor found some unrealistic patients in it. Please see the overall comment section above for more information.
>
> Question 2: We haven’t had the opportunity to modify / fine tune the architecture of the classifier network.
>
> Question 3: We haven’t yet considered this case. We feel that, before tackling this issue, we first needed to tackle another problem that was deemed very important by the doctor who helped us. We needed to deal with the disconnect between the problem definition used by the machine learning community and the problem definition as considered by the medical community. The first RL agent that we built used a SymCat-based dataset and was trained as most automatic diagnosis systems, to predict the single pathology that the patient was suffering from, while asking as few questions as possible to avoid burdening the patient. We asked our collaborating doctor to analyze the interactions between this agent and the synthetic patients. The doctor was extremely disappointed and indicated that the system was “cutting corners” by asking too few questions that weren’t medically sound. Moreover, the agent was trained to predict a single pathology, while doctors reason in terms of a differential diagnosis, and the agent wasn’t paying enough attention to severe pathologies. The doctor indicated that physicians wouldn’t use such a tool because they wouldn’t trust it. It became clear at this point that we needed to work with the doctor to refine the problem definition and try to share what we learned with the machine learning community. We view our proposed approach as an initial step in terms of properly defining the problem, as well as defining an initial set of metrics that can assess the performance of a model. More work is needed in this direction and it is paramount that this work be done with the medical community.
> As for the patient answers, there are at least 4 aspects that need to be considered: 1) patients not knowing what to answer (e.g., “I am not sure” or “I don’t know”), 2) patients being imprecise about their answer (e.g., indicating that the pain intensity is 4/10 when it’s actually equal to 5/10), 3) patients having different perceptions of their symptoms (e.g., some patients have higher tolerance to pain than other patients, and might provide different answers), and 4) patients who might not be fully truthful and transparent in their answers (e.g., patients who purposely provide incorrect answers). We would like to explore all 4 aspects in future work. Some authors have already explored the first aspect by replacing some patient answers with a “I am not sure” answer.

---

### Official Review · Reviewer_XzH6 · 2022-07-11

**Rating:** 4
**Confidence:** 4
**Ethics Flag:** Yes
**Soundness:** 3 good
**Presentation:** 3 good
**Contribution:** 3 good

**Summary:**

The paper focuses on the automatic diagnosis task and the authors reformulate the task of evidence acquisition and automated diagnosis by introducing doctors’ trust as a desideratum. The authors propose a new RL model (i.e., CASANDE) with DQN as the backbone and mimic human doctors’ reasoning by incorporating the exploration-confirmation and ruling out severe pathologies with new reward functions. The experiments on a synthetic dataset show that the proposed model outperforms the baselines.

**Questions:**

The authors will at least need to address questions 1, 2, 3, and 5 in the Weaknesses section.

**Ethics Review Area:**

["Inadequate Data and Algorithm Evaluation", "Responsible Research Practice (e.g., IRB, documentation, research ethics)"]

**Limitations:**

Yes. The section "Limitations and potential negative social impact" is sufficient.

**Strengths And Weaknesses:**

Strengths:
1. The authors propose a new RL model, CASANDE, to interact with patients and produce differential diagnoses automatically. The task is significant in the clinical setting and could reduce human doctors’ workload.
2. The authors mimic human doctors’ reasoning, namely using differential diagnosis with the exploration-confirmation approach while prioritizing severe pathologies, by designing new reward functions, which is helpful for convincing doctors and real-world deployment.
3. The authors conduct experiments on a synthetic dataset and the results show that the proposed model outperforms state-of-the-art methods.

Weaknesses:
1. The proposed model is similar to an RL baseline AARLC. The main difference is that the proposed model has four additional reward components. Table 2 shows that after removing the four components, the performance is still much better than AARLC in Table 1. It is worth more discussion and explanation.
2. The paper’s main contribution is the four reward components in Table 2. However, the performance difference among the proposed model's various versions is insignificant (i.e., about 1% improvement on DDF1, 0.5% improvement on DSF1 and PER, and worse performance on IL).
3. The expense of misdiagnosis is much more expensive than the cost of interaction with patients in clinical settings. It is unclear how the authors balance the two kinds of expenses. It would be better if the authors conducted experiments to show the automatic diagnosis performance over different lengths of interaction with patients.
4. The patients in the dataset are simulated (including the demographics, symptoms, and diagnosis). The experimental results on a totally synthetic dataset are not convincing enough.
5. It would be better if the authors displayed the standard deviation in Table 2.
6. It would be better if the authors gave some case studies to show how the proposed model is beneficial from designed reward functions.

---

> ### Author Response · Authors · 2022-08-02
> **Feedback to reviewer XzH6**
>
> Weakness 1: We recently made additional changes to AARLC to better handle the differential diagnosis. Those changes, which are described in section B.4 of the supplementary, led to improved AARLC performance. We updated Table 1, Figures 2 and 3 to reflect those changes.
> There are several differences between AARLC and CASANDE that are not limited to the reward functions: 1) AARLC uses separate models for the classifier and policy networks while in CASANDE, those 2 networks share the same encoder and exchange information. 2) The classifier network is updated at each interaction turn in AARLC. In CASANDE, it is only updated at the end of the interaction as this is when the differential diagnosis is needed and can be accurately predicted. Forcing the classifier to predict the ground truth differential at each turn can confuse the classifier, in particular at the start of the interaction, when the number of collected evidence is small (e.g., if a patient indicates that they are coughing, there isn’t enough information to predict the differential). 3) AARLC provides a small positive reward when the agent asks questions about evidence that the patient doesn’t have. CASANDE doesn’t because it isn’t clear when questions about negative evidence are useful: some questions can be useful because they can rule out several pathologies from the differential; others are not informative. It might be worth exploring in future work a reward about negative evidence that is based on the impact of this information on the differential. 4)  AARLC has the r_H reward which encourages the model to reduce the entropy of the differential from one interaction turn to the next, as more information is accumulated. In CASANDE, we use another strategy (in the consolidation phase), which consists in ensuring that the differential at the next interaction turn is closer to the ground truth differential than at the previous turn. In a sense, we consider that it is not sufficient to reduce the entropy of the differential if the differential is not being pushed in the right direction. 5) AARLC’s policy network doesn’t predict an exit action; instead, AARLC compares the entropy of the differential at each interaction turn to a learnable threshold and stops the interaction when the entropy is smaller than the threshold, irrespective of whether the differential is correct. CASANDE’s policy network can directly predict the exit action, and its loss (Eq. 8) provides feedback to the network at each interaction turn about the value of predicting the exit action; it also has a reward at the end of the interaction that depends on the quality of the predicted different diagnosis.
>
> Weakness 2: We think that our paper has several contributions: 1) Paving the way for a better problem definition for automatic diagnosis and evidence collection systems based on a collaboration with a doctor (whose feedback was corroborated by scientific papers and by other doctors to whom we recently presented our work). We reformulated the problem to better align the ML and medical communities, and made initial attempts to develop a model and design evaluation metrics towards that goal. Much more work is needed before a system can be deployed. But success cannot be achieved if ML researchers don’t collaborate with medical experts. 2) Besides the 4 reward functions, there are several aspects in CASANDE that help it achieve better performance (such as the aforementioned elements). DDF1, DSF1 and PER are statistically better than other approaches even when the 4 reward functions are disabled. 3) As to the ablation study, the +0.5% improvement in DSF1 might seem small, but for patients suffering from severe pathologies, this can be a life and death situation. Similarly, an increased interaction length of 2 turns for a +1% improvement in DDF1 might lead to a better follow up for patients.
>
> Weakness 3: Instead of trying different interaction lengths, we set the maximum length in our experiments to be large (30). Based on the results in Table 1, the average interaction length is ~20 turns and thus much smaller. It’s therefore unclear whether having a larger upper bound will improve the predicted differential. We feel that it would be more beneficial to have a doctor analyze the errors made by the agent to better plan for future improvements.
>
> Weakness 4: There are 2 real-patient datasets (DX and MuZhi), which don’t contain differentials and are too small to train an RL agent (as pointed out in other papers).
> Although we’re using a fully synthetic dataset, we worked with a doctor to validate our work, except for the latest version of CASANDE which the doctor hasn’t yet had the time to check. Several qualitative evaluations were made by the doctor during the project. Those are described in the overall response section (please see the "General response to all reviewers" ).
>
> Weakness 5: We updated Table 2 and added Figure 10.
>
> Weakness 6: We added examples in the appendix, section E.

---

> > ### Comment · Reviewer_XzH6 · 2022-08-03
> > **Quick questions**
> >
> > Thanks for your reply. Skimming through the response, I have two additional quick questions:
> > - For weakness 3, is it easy "to show the automatic diagnosis performance over different lengths of interaction with patients" e.g., <10, 10-20, > 20?
> > - Are the confidence intervals in Table 1 and Table 2 inconsistent? For example, In table 1, IL: 0.46, DDF1: 0.55, DSF1: 0.34, PER: 0.86; but in table 2, IL: 0.18, DDF1: 0.24, DSF1: 0.13, PER: 0.35.

---

> > > ### Author Response · Authors · 2022-08-05
> > > **Response to quick questions**
> > >
> > > Thanks for the follow-up questions.
> > >
> > >
> > > ## Response to question 1
> > >
> > > The performance for the three groups is shown below. Values are mean of 3 runs and 95% confidence intervals are in brackets. Bold fonts indicate best values, and italic indicates values insignificantly different from best values. We are also ready to present more fine-grained results in the camera-ready version.
> > >
> > > | IL | $(1,10]$ | $(10,20]$ | $(20,30]$ |
> > > |:---:|:---:|:---:|:---:|
> > > | PER | 90.31 (20.50) | 98.57 (0.88) | **99.16 (0.36)** |
> > > | DDF1 | 90.36 (7.57) | _94.45 (0.82)_ | **94.47 (0.16)** |
> > > | DDP | 87.00 (13.85) | **92.45 (1.74)** | _92.30 (0.93)_ |
> > > | DDR | **98.45 (2.84)** | _98.24 (0.60)_ | _98.06 (0.84)_ |
> > > | DSF1 | 60.20 (23.93) | _71.12 (4.66)_ | **77.30 (6.99)** |
> > > | DSP | **98.15 (2.02)** | _97.83 (0.50)_ | _96.96 (0.29)_ |
> > > | DSR | 60.98 (25.32) | _71.77 (4.84)_ | **78.35 (7.02)** |
> > > | Number of ground truth evidences | 4.81 (1.48) | 10.85 (0.40) | 16.56 (0.16) |
> > > | Size of ground truth differentials | 2.37 (1.20) | 8.20 (0.32) | 10.02 (0.42) |
> > >
> > > We argue that the benefit of having longer interactions is a complicated question. First, we observe that some metrics improve with longer interactions such as PER, DDF1, DSF1 and DSR, however the improvement is marginal or insignificant between groups of 10 < IL<= 20 and IL > 20. This suggests that further increasing the maximal turn may have diminishing returns.
> > >
> > > Second, the patients with whom the agent has shorter interactions may be different from those with whom the agent has longer interactions. In fact, IL is highly correlated with the number of ground truth evidences and the size of the ground truth differentials (Pearson correlation of 0.97 and 0.95 respectively). Thus, we should not consider the metrics for the three groups uniformly, but rather take into account that the difference may come from the difference in patients.
> > >
> > > Finally, we want to point out that only 1.2% interactions are shorter than 10 turns and 0.5% interactions use the maximal 30 turns.
> > >
> > > ## Response to question 2
> > >
> > > The numbers in Table 2 that you are referring to are standard deviations, not confidence intervals. In table 2, the numbers in **brackets** are confidence intervals and the numbers after $\pm$ are standard deviations.
> > >
> > > Still, we notice there is a small discrepancy between the values in Table 2 and in Table 1 (e.g. a difference of 0.01 for IL’s confidence interval). We found that the difference is due to Google sheet’s automatic rounding behavior, which we used for calculating means, CIs and stds for Table 2 but not for Table 1. We recalculated Table 2 in Python, as we did for Table 1, and confirmed that the values are identical. Importantly, the significant levels and our conclusions remain unchanged.
> > >
> > > We include the updated Table 2 below, and we are ready to correct it in the camera-ready version.
> > >
> > > | R_ex | R_co | R_sev | R_cl | IL | DDF1 | DSF1 | PER |
> > > |:---:|:---:|:---:|:---:|:---:|:---:|:---:|:---:|
> > > | ✕ | ✕ | ✕ | ✕ | 17.56 (0.25) ± 0.06 | 93.11 (0.70) ± 0.16 | 73.55 (0.27) ± 0.06 | 98.24 (0.36) ± 0.08 |
> > > | ✕ | ✕ | ✕ | ✓ | 17.79 (0.13) ± 0.03 | 93.48 (1.31) ± 0.30 | 73.33 (0.28) ± 0.06 | 98.32 (0.88) ± 0.20 |
> > > | ✓ | ✓ | ✓ | ✕ | 19.09 (0.38) ± 0.09 | 94.26 (2.82) ± 0.65 | 74.19 (0.41) ± 0.10 | 98.34 (1.13) ± 0.26 |
> > > | ✕ | ✓ | ✓ | ✓ | 18.01 (0.36) ± 0.08 | 94.49 (0.27) ± 0.06 | 73.92 (0.38) ± 0.09 | 98.48 (1.10) ± 0.26 |
> > > | ✓ | ✕ | ✓ | ✓ | 18.91 (1.36) ± 0.32 | 94.16 (0.69) ± 0.16 | 73.86 (0.34) ± 0.08 | 98.54 (0.68) ± 0.16 |
> > > | ✓ | ✓ | ✓ | ✓ | 19.71 (0.46) ± 0.11 | 94.24 (0.55) ± 0.13 | 73.88 (0.34) ± 0.08 | 98.39 (0.86) ± 0.20 |

---

> > > > ### Comment · Reviewer_XzH6 · 2022-08-05
> > > > **Thank you for the reply.**
> > > >
> > > > Yes. Tables 1 and 2 are consistent. Thank you.
> > > >
> > > > I suggest the authors compute the metrics for the same patients with less interaction (e.g., 10) and more interactions (e.g, 30), then report the results. In this way, it would be interesting whether the number of interactions can significantly affect the performance.

---

> > > > > ### Author Response · Authors · 2022-08-09
> > > > > **Response to follow-up suggestion**
> > > > >
> > > > > Thank you for the suggestion. We conducted several follow-up experiments following our understanding of the suggestion.
> > > > >
> > > > > First, we force CASANDE (trained with maximally allowed 30 turns) to always interact for a fixed amount of turns at test time, for instance 10, 20, and 30 (maximally allowed interaction length), and evaluate the performance. The results (mean and 95% CI) are shown below:
> > > > >
> > > > > |  | Force to 10 turns | Force to 20 turns | Force to 30 turns |
> > > > > |:---:|:---:|:---:|:---:|
> > > > > | PER | 93.30 (15.19) | 99.22 (0.97) | 99.49 (0.65) |
> > > > > | DDF1 | 89.78 (9.37) | 94.34 (1.56) | 93.12 (1.85) |
> > > > > | DDP | 85.87 (15.87) | 92.39 (2.89) | 91.00 (3.32) |
> > > > > | DDR | 98.64 (2.08) | 98.13 (0.64) | 97.44 (0.77) |
> > > > > | DSF1 | 60.32 (23.47) | 70.85 (4.78) | 72.76 (0.30) |
> > > > > | DSP | 97.99 (2.12) | 97.84 (0.70) | 97.12 (0.69) |
> > > > > | DSR | 61.15 (24.69) | 71.45 (5.00) | 73.57 (0.43) |
> > > > >
> > > > > If CASANDE is forced to interact for 10 turns, there’s a decrease in performance; however the decrease is not statistically significant due to high variance across 3 runs. At the very least, this suggests that 10 turns is too few for CASANDE to work properly. If CASANDE is forced to interact for 20 turns or for the full 30 turns, the change in performance is marginal or insignificant. This shows that 30 turns is sufficient for our experiments, and forcing CASANDE to increase interaction length does not significantly affect performance.
> > > > >
> > > > > Second, we also trained CASANDE with a maximum of 10 turns and evaluated its test performance. The results (average and 95% CI) are shown below, along with the results of the original CASANDE from Table 1, trained with a maximum  of 30 turns:
> > > > >
> > > > > |  | IL (including initial evidence) | DDF1 | DSF1 | PER |
> > > > > |:---:|:---:|:---:|:---:|:---:|
> > > > > | Trained up to 10 turns | 10.85 (0.05) | 84.85 (0.77) | 69.03 (0.31) | 71.84 (0.65) |
> > > > > | Trained up to 30 turns | 19.71 (0.46) | 94.24 (0.55) | 73.88  (0.34) | 98.39 (0.86) |
> > > > >
> > > > > The performance decreases significantly compared to allowing up to 30 turns. However, this is expected as 10 turns should not be enough for uncovering all positive evidences. This can be observed in the difference in IL: the model tends to use all available turns when trained with up to 10 turns (mean IL is close to the upper limit), but that is not the case when it is trained with up to 30 turns (mean IL is much lower than the upper limit).
> > > > >
> > > > > In summary, while using too few turns might adversely affect performance, the number of turns allowed in our experiments is sufficient, and prolonging interactions does not significantly affect performance. Finally, we would like to reiterate that the impact of interaction length is a complicated question and should be considered with patient’s characteristics in mind.

---

### Official Review · Reviewer_hhxa · 2022-07-13

**Rating:** 6
**Confidence:** 2
**Soundness:** 3 good
**Presentation:** 3 good
**Contribution:** 3 good

**Summary:**

An interesting paper which focuses on imitating doctor's reasoning in automated clinical diagnosis tasks. As per the authors, the paper proposes a novel reinforcement learning (RL) agent to this end, and utilises it to model the evidence acquisition and automatic
diagnosis tasks. The proposed RL agent outperforms multiple baseline models based on experiments with benchmark datasets.

**Questions:**

- Did the authors perform a qualitative evaluation of the model's performance by discussing the predictions/outputs with domain-experts (real-world doctors) etc.? Besides the comparison on the benchmark datasets - how trustworthy/transparent/explainable is the proposed DRL framework for real-world utility - please clarify.

UPDATE: Thanks for addressing this reviewer's concerns. I have increased the score from 6 to 7.

**Limitations:**

Limitations have been properly addressed in the paper.

**Strengths And Weaknesses:**

The paper has a real-world application and the approach can be quite impactful in safety critical applications like clinical diagnosis
However, there are few things which reduce the overall merit of the paper:
- The results and experiments provide an evaluation of the model based on metrics like F1 scores etc. (e.g. in Figure 3) which is useful. However, it is still not entirely clear for an average. reader in the AI community (who may not be specialised in the domain of DRL) that how exactly is the doctor's reason emulated by the model? It would have been useful if the authors provide some example(s) to describe the medical diagnosis reasoning imitation by the proposed DRL framework in comparison to a medical doctor. Can it be done with e.g. a visual depiction of the model's outputs/predictions? Figure 1 shows the proposed framework and the example of the synthetic interaction with the proposed model is useful - but it does not resonate clearly with the results. It is suggested that a better representation of the final model outputs/predictions later on in the paper's results section (besides the quantitative evaluation would be useful).

---

> ### Author Response · Authors · 2022-08-02
> **Feedback to reviewer 2 and all reviewers (continuing from general response)**
>
> - Weakness 1:
>
> We added examples of interactions between CASANDE and synthetic patients in the appendix, section E. We also present more details about the ablation study: Table 2 in the main text is now augmented with Figure 10 in the appendix.
>
> - Question 1:
>
> Several qualitative evaluations were made by a doctor during the project. Those are described in the overall response section (please see the top most comment section / General response to all reviewers).  But the doctor hasn’t done a qualitative evaluation of the recent version of CASANDE due his workload at the clinic and hospital where he works, and due to family considerations. We don’t yet know when he will be able to evaluate the latest results.
>
> With respect to the “how trustworthy/transparent/explainable is the proposed DRL framework for real-world utility” question:
> CASANDE can provide the following information at each interaction turn:
> Question and patient answer.
> Exploration score that can guide the doctor in understanding whether CASANDE is still analyzing different scenarios.
> Confirmation score that can guide the doctor in understanding whether CASANDE is trying to consolidate its differential.
> Current differential diagnosis.
> At the end of the interaction, it provides the final differential diagnosis.
>
> We don’t know whether this information is sufficient for all doctors. It is for the doctor with whom we’ve been working. But this doctor is a bit an “outlier” because he is interested in and understands the basics of machine learning. Further discussions with the medical community are needed.
>
> **DOCTOR EVALUATION RESULTS (CONTINUING FROM GENERAL RESPONSE)**
>
> The doctor defined the following evaluation criteria, with a score on a 5-point Likert scale:
> - Q1: The agent asks relevant questions.
> - Q2: The questions asked allow me to establish a differential diagnosis.
> - Q3: The agent asks enough questions to make a differential diagnosis.
> - Q4: The questions asked are similar to what I would have asked.
> - Q5: The information collected is useful for me to continue assessing the patient.
> - Q6: The sequence of questions seems logical to me.
>
> The Likert scale is:
> 1. strongly disagree
> 2. disagree
> 3. neutral
> 4. agree
> 5. strongly agree
>
> Following are the scores (note that one doctor forgot to evaluate 2 patients):
>
> | Patient | Doctor | Q1 | Q2 | Q3 | Q4 | Q5 | Q6 |
> |:-------:|:------:|:--:|:--:|:--:|:--:|:--:|:--:|
> |    1    |    1   |  4 |  2 |  2 |  4 |  4 |  3 |
> |         |    2   |  4 |  4 |  2 |  2 |  4 |  2 |
> |         |    3   |  4 |  4 |  3 |  4 |  4 |  4 |
> |    2    |    1   |  4 |  5 |  4 |  4 |  4 |  4 |
> |         |    2   |  5 |  5 |  5 |  4 |  5 |  5 |
> |         |    3   |  4 |  4 |  4 |  3 |  4 |  4 |
> |    3    |    1   |  4 |  4 |  3 |  4 |  4 |  4 |
> |         |    2   |  5 |  5 |  5 |  4 |  5 |  5 |
> |         |    3   |  5 |  5 |  5 |  5 |  5 |  5 |
> |    4    |    1   |  5 |  4 |  5 |  4 |  5 |  5 |
> |         |    2   |  5 |  5 |  5 |  4 |  5 |  5 |
> |         |    3   |  5 |  4 |  5 |  4 |  5 |  5 |
> |    5    |    1   |  4 |  5 |  4 |  4 |  5 |  4 |
> |         |    2   |  2 |  3 |  2 |  2 |  3 |  2 |
> |         |    3   |  4 |  5 |  4 |  4 |  5 |  4 |
> |    6    |    1   |  4 |  4 |  4 |  4 |  4 |  3 |
> |         |    2   |  2 |  3 |  4 |  2 |  4 |  1 |
> |         |    3   |  4 |  4 |  4 |  3 |  4 |  2 |
> |    7    |    1   |  4 |  5 |  4 |  3 |  4 |  3 |
> |         |    2   |  3 |  3 |  3 |  2 |  4 |  2 |
> |         |    3   |  4 |  4 |  4 |  4 |  4 |  3 |
> |    8    |    1   |  3 |  4 |  4 |  2 |  4 |  3 |
> |         |    2   |  - |  - |  - |  - |  - |  - |
> |         |    3   |  4 |  4 |  4 |  4 |  4 |  4 |
> |    9    |    1   |  4 |  4 |  4 |  3 |  4 |  4 |
> |         |    2   |  - |  - |  - |  - |  - |  - |
> |         |    3   |  4 |  4 |  3 |  4 |  4 |  4 |
> |    10   |    1   |  4 |  4 |  4 |  4 |  4 |  4 |
> |         |    2   |  5 |  5 |  5 |  5 |  5 |  5 |
> |         |    3   |  5 |  4 |  5 |  4 |  5 |  5 |

---

> > ### Comment · Reviewer_hhxa · 2022-08-08
> > **Thanks for your response**
> >
> > UPDATE: Thanks for addressing this reviewer's concerns. I have increased the score from 6 to 7.

---

### Official Review · Reviewer_Ynqo · 2022-07-18

**Rating:** 7
**Confidence:** 4
**Soundness:** 3 good
**Presentation:** 3 good
**Contribution:** 3 good

**Summary:**

The paper moves beyond the traditional way of improving the prediction accuracy of the patient’s pathology. It enables the system to optimize with the differential diagnosis with the exploration-confirmation approach while prioritizing severe pathologies. The proposed methodology performed well on benchmark datasets.

**Questions:**

1. Eqn 1, it is unclear why another divergence or distance was not used. Might not we consider KL or the earth-mover's in place of JSD. A small discussion behind the justification of taking JSD would help the paper to understand more.
2. Eqn 2, the weight is learned, but what if the dataset has some bias, such as gender, race, belief, etc.? Also, if we say the end decision would be taken by humans, aren't we encouraging human bias in the systems too!
3. Eqn 5. what are the values of $\gamma$ for this study? Do we have any ablation study on the same - or am I missing something here.
4. It seems like the work is close to the work [1]

## Ref:
--------------
[1] Slice-based Learning: A Programming Model for Residual Learning in Critical Data Slices. Vincent S. Chen et al. https://vincentsc.com/papers/chen2019slice.pdf


**Limitations:**

Justified clearly

**Strengths And Weaknesses:**

### Strength
-------------------------
1. Reproducibility: I liked how the authors provided and documented the code well. Some code snippets are $a \ little$ hard to follow, but I am confident that the authors would clean them at the open source time.
2. Figures and tables: Starting from the motivational diagram to the supl. material - the authors provided good figures that helped me understand the paper's central theme and the texts.
3. Formulation: the equations are easy to follow and supported with adequate description - making them easy to read and criticize.


### Weaknesses
-------------------------
1. Presentation style of the paper is relatively not very impressive. A few suggestions could be; (1) merging the Intro and the Medical history taking, and (2) the preliminaries could be merged with the method.
2. A few mathematical formulations are used without proper justification; pls see Sec. Questions.

---

> ### Author Response · Authors · 2022-08-02
> **Feedback to reviewer Ynqo**
>
> Weakness 1: Thank you for this feedback. We will rework those sections to improve the clarity and flow of information.
>
> Question 1: Different distance functions could have been used. We initially considered using the KL function but, following discussions with a doctor, we chose to use JSD because it is symmetrical. The symmetry property of JSD is useful because we are interested in verifying whether the differential diagnoses of two successive steps are different without assuming some kind of directionality (based on the fact that the doctor who helped us on this project indicated that he initially explores different scenarios and doesn’t directly zero-in on a specific differential). KL assumes a directionality (by measuring the divergence of one probability distribution from another distribution). We haven’t run experiments with other distance functions. This could be done as future work, if we can identify relevant functions.
>
> Question 2: There are two kinds of biases, one is the inductive bias we want the model to have, the other is the kind that reflects social discriminations which we want to avoid. A main focus of this work is to encourage the first kind of bias in the model to emulate doctors and their domain expertise. For the second kind, the potential risk of bias applies to any dataset (and to all the weights of the models using the dataset). What we can do is first use the dataset that makes the effort to reflect socio-demographics and medical characteristics fairly. Second, we emphasize that the models always need to be retrained considering the socio-demographics and medical characteristics of any specific population before applying them to this population. Third, prior to deploying any model in a production environment, there is a need to develop a rigorous experimental protocol to validate the model; this work requires a close collaboration with the medical community, and possibly government agencies when official certifications are required.
>
> In terms of the end decision being taken by humans, this is currently important and probably necessary because, based on our knowledge, there aren’t legal frameworks governing the responsibility of fully automated diagnosis systems that operate without human intervention. Moreover, the medical community doesn’t seem ready for  the integration of such systems. In our work, based on our discussion with medical experts, we’ve been assuming that the goal is to build a tool that can support doctors by alleviating their workload, and not replace them. Switching to fully automated systems will require building the trust of the medical community by working closely with it and building models that account for elements which are medically important.
>
> Question 3: \gamma is the discount factor used in reinforcement learning and is set to 0.99. This is a standard value used in many RL papers. We haven’t tried to fine tune it.
>
> Question 4: We checked the “Slice-based learning” paper. It is unclear to us how our work is close to the paper. Can you please provide more details? Based on our understanding, this paper proposes an approach to train models that have a good overall performance on a specific task/data as well as a good performance on specific subsets of the data which are deemed important. For example, when building an autonomous driving system, it is important to make sure that this system behaves properly all day long, including at night time. When building an NLU system, the model should behave properly on all passages, including long ones, etc. Was this reference shared with us because we care about properly handling severe pathologies, which are a subset of the overall set of pathologies? If so, the way we handle several pathologies is very different from the slice-based learning paper: we use reward functions based on the rule-in and rule-out rate of severe pathologies from the differential diagnosis. The slice-based learning approach requires splitting the data into slices by defining slice functions. We haven’t considered such an option. It could be worthwhile to see if such an approach can be used to improve the performance of our RL agent on severe pathologies.

---

### Review · Ethics_Reviewer_zKZp · 2022-08-05

**Recommendation:**

I strongly suggest including a discussion of ethical concerns, as detailed above in the Ethics Review.

**Ethical Issues:**

Yes

**Ethics Review:**

The paper raises several ethical issues, but I believe that they can be addressed by discussing the limitations and potential negative social impact of the work in more detail.

My primary concern is related to the potential for bias and discrimination. Much research has documented evidence of race and gender based discrimination in health care (e.g., [1,2]). In recent years much work has studied the problem of the emergence of discrimination in machine learning models, commonly referred to as algorithmic fairness (please refer to [3] for a review of recent work).

Given that this paper focuses on a sensitive and societally consequential domain with a documented history of issues related to discrimination, it is of outmost importance to thoroughly study the fairness of any algorithm which may be utilized in such settings. While it is not realistic to expect the authors to conduct such a study, I believe that it is crucial for the authors to emphasize that before deploying such a tool, it is necessary to (i) rigorously evaluate its fairness, and (ii) employ the state-of-the-art techniques for mitigating unfairness in machine learning models.

Additionally, I believe that it should be emphasized that prior to the deployment of such algorithms in practice, it is necessary to (i) rigorously evaluate their predictive performance on real-world datasets (unlike the synthetic dataset utilized here), and to (ii) conduct human-subject experiments to study how both doctors and patients would utilize such tools.

* [1] Shavers, Vickie L., et al. "The state of research on racial/ethnic discrimination in the receipt of health care." American journal of public health 102.5 (2012): 953-966.
* [2] Puhl, Rebecca M., Tatiana Andreyeva, and Kelly D. Brownell. "Perceptions of weight discrimination: prevalence and comparison to race and gender discrimination in America." International journal of obesity 32.6 (2008): 992-1000.
* [3] Corbett-Davies, Sam, and Sharad Goel. "The Measure and Mismeasure of Fairness: A Critical Review of Fair Machine Learning." (2018).

---

### Review · Ethics_Reviewer_E5WA · 2022-08-09

**Recommendation:**

I recommend the authors describe the consultation process with the collaborating physicians in greater detail in the appendix. One thing that I find less satisfactory about this study is there is no empirical evidence to test whether doctors trust this model more so than other models. Theoretically, the model described in the paper has trustworthiness properties, but ultimately, trust is subjective. Some HCI types of testing will improve the arguments that the authors' model will generate more trust among physicians.

**Ethical Issues:**

Yes

**Ethics Review:**

The paper raises a number of ethical issues including:
1) doctors in different locations and cultures may be affected by other factors when deciding to trust the algorithm
2) synthetic patient data may not represent real patients
3) patients could overly trust the automated diagnosis system when it's no substitute for human doctors

---

### Author Response · Authors · 2022-08-02
**General response to all reviewers**

Based on the reviewers' questions, we would like to explain how the proposed approach came about. Several qualitative evaluations were made during the project by a doctor, except for the final model because the doctor hasn’t been able to help us recently due to his workload and family considerations. Following is a summary of his evaluations:
1. In our first RL agent, we mimicked the base techniques proposed in papers published by the ML community, and whose goals are to predict the patient’s underlying pathology while asking as few questions as possible. This agent was built using a synthetic dataset derived from the SymCat medical knowledge base. The doctor verified interactions between the agent and some synthetic patients, and was very disappointed. He indicated that the agent was “cutting corners” by asking too few questions, often leading to an incomplete medical reasoning. He said that for him to trust such a tool, the tool would need to consider important aspects of a doctor’s reasoning. Those include: 1) Generating a differential diagnosis, as it is sometimes impossible to identify the pathology the patient is suffering from just by inquiring about the patient’s symptoms/antecedents. 2) A doctor needs to pay special attention to severe pathologies as a mistake in handling such pathologies can lead to major repercussions for patients. 3) A doctor explores different scenarios at the beginning of the interaction and doesn’t immediately zero-in on a differential. He then moves on to ask questions to confirm he has the right differential. 2. Based on this feedback (which was recently validated by other doctors to whom we presented our work), it became clear the problem definition used by ML researchers needed to be refined. The doctor’s feedback helped us improve this definition. We do not claim that this work is over. We have started paving the way for a better definition and think that more work with the medical community is needed.
2. The doctor analyzed our SymCat-based synthetic dataset. He was worried that it didn’t include differential diagnoses and disease severity information. He was also worried about the “realism” of some of the synthetic patients. He identified incorrect statistics in SymCat and found some symptom definitions to be incorrect.
3. Based on the doctor’s feedback, we decided to build an RL agent that takes into consideration the elements deemed important by the doctor. We first asked the doctor to check the DDXPlus dataset. While the dataset isn’t perfect, the  doctor was much more satisfied with its quality, which he considered good enough to build a proof of concept. We then proceeded to build an initial version of CASANDE, which is similar to the one presented in our paper, but with some differences: 1) our previous agent didn’t handle severe pathologies; 2) we had a negative reward for each question about an evidence the patient wasn’t experiencing; 3) the classifier branch was updated at each interaction turn instead of only being updated at the last turn; 4)  the agent wasn’t using the hierarchy symptom information found in DDXPlus and was sometimes asking questions in an order that wasn’t logical. The doctor analyzed interactions between this agent and some synthetic patients and indicated that he was quite happy.
4. Based on this earlier CASANDE version, the doctor and two of his colleagues (who weren’t involved in the project) analyzed the interactions of the agent with 10 synthetic patients who were randomly selected from the DDXPlus test set.

**PLEASE REFER TO OUR RESPONSE TO REVIEWER HHXA FOR THE CONTINUATION OF OUR GENERAL RESPONSE**

---

### Meta-Review · Area_Chair_ydhn · 2022-08-23

**Recommendation:** Accept
**Confidence:** Less certain

**Metareview:**

**Technical Review and Decision**: This paper proposes four sets of rewards such that an RL maximizing the sum of those rewards (combined with the environment reward) to mimic doctors' behavior and increase the trust in the automatic differential diagnosis systems. The paper is well-written and the exchange between the reviewers and the authors have been constructive. There are multiple questions and the reviewers are convinced by the response. The authors should include the clarifications in the camera-ready version of the paper. While the methodological contributions are limited to a reward design, this paper qualifies as a good application paper.

**Ethics Review**: The ethical reviewers have identified that the authors need to elaborate more on the doctor consultation process and make it more transparent. I strongly suggest including a discussion of ethical concerns, as discussed in the ethical reviews.

**Award:**

No

---

### Decision · Program_Chairs · 2022-09-14

Accept